# Hospital Variation in Cancer Treatments and Survival OutComes of Advanced Melanoma Patients: Nationwide Quality Assurance in The Netherlands

**DOI:** 10.3390/cancers13205077

**Published:** 2021-10-11

**Authors:** Jesper van Breeschoten, Alfonsus J. M. van den Eertwegh, Liesbeth C. de Wreede, Doranne L. Hilarius, Erik W. van Zwet, John B. Haanen, Christian U. Blank, Maureen J. B. Aarts, Franchette W. P. J. van den Berkmortel, Jan Willem B. de Groot, Geke A. P. Hospers, Ellen Kapiteijn, Djura Piersma, Rozemarijn S. van Rijn, Marion A. M. Stevense-den Boer, Astrid A. M. van der Veldt, Gerard Vreugdenhil, Marye J. Boers-Sonderen, Karijn P. M. Suijkerbuijk, Michel W. J. M. Wouters

**Affiliations:** 1Dutch Institute for Clinical Auditing, Rijnsburgerweg 10, 2333 AA Leiden, The Netherlands; j.vanbreeschoten@dica.nl; 2Department of Medical Oncology, Amsterdam UMC, VU University Medical Center, Cancer Center Amsterdam, De Boelelaan 1118, 1081 HZ Amsterdam, The Netherlands; vandeneertwegh@amsterdamumc.nl; 3Department of Biomedical Data Sciences, Leiden University Medical Centre, Einthovenweg 20, 2333 ZC Leiden, The Netherlands; L.C.de_Wreede@lumc.nl (L.C.d.W.); e.w.van_zwet@lumc.nl (E.W.v.Z.); 4Department of Pharmacy, Rode Kruis Ziekenhuis, Vondellaan 13, 1942 LE Beverwijk, The Netherlands; dhilarius@rkz.nl; 5Department of Medical Oncology and Immunology, Netherlands Cancer Institute, Plesmanlaan 121, 1066 CX Amsterdam, The Netherlands; j.haanen@nki.nl (J.B.H.); c.blank@nki.nl (C.U.B.); 6Division of Molecular Oncology & Immunology, Netherlands Cancer Institute, Plesmanlaan 121, 1066 CX Amsterdam, The Netherlands; 7Department of Medical Oncology, GROW School of Oncology and Developmental Biology, Maastricht University Medical Centre+, P. Debyelaan 25, 6229 HX Maastricht, The Netherlands; mjb.essers.aarts@mumc.nl; 8Department of Medical Oncology, Zuyderland Medical Centre Sittard, Dr. H. van der Hoffplein 1, 6162 BG Sittard-Geleen, The Netherlands; f.vandenberkmortel@zuyderland.nl; 9Isala Oncology Center, Isala, Dokter van Heesweg 2, 8025 AB Zwolle, The Netherlands; j.w.b.de.groot@isala.nl; 10Department of Medical Oncology, University Medical Centre Groningen, University of Groningen, Hanzeplein 1, 9713 GZ Groningen, The Netherlands; g.a.p.hospers@umcg.nl; 11Department of Medical Oncology, Leiden University Medical Centre, Albinusdreef 2, 2333 ZA Leiden, The Netherlands; h.w.kapiteijn@lumc.nl; 12Department of Internal Medicine, Medisch Spectrum Twente, Koningsplein 1, 7512 KZ Enschede, The Netherlands; D.Piersma@mst.nl; 13Department of Internal Medicine, Medical Centre Leeuwarden, Henri Dunantweg 2, 8934 AD Leeuwarden, The Netherlands; rozemarijn.van.rijn@znb.nl; 14Department of Internal Medicine, Amphia Hospital, Molengracht 21, 4818 CK Breda, The Netherlands; MStevense@amphia.nl; 15Department of Medical Oncology and Radiology & Nuclear Medicine, Erasmus Medical Centre, ‘s-Gravendijkwal 230, 3015 CE Rotterdam, The Netherlands; a.vanderveldt@erasmusmc.nl; 16Department of Internal Medicine, Maxima Medical Centre, De Run 4600, 5504 DB Eindhoven, The Netherlands; G.Vreugdenhil@mmc.nl; 17Department of Medical Oncology, Radboud University Medical Centre, Geert Grooteplein Zuid 10, 6525 GA Nijmegen, The Netherlands; Marye.Boers-Sonderen@radboudumc.nl; 18Department of Medical Oncology, University Medical Centre Utrecht, Heidelberglaan 100, 3584 CX Utrecht, The Netherlands; K.Suijkerbuijk@umcutrecht.nl; 19Department of Surgical Oncology, Netherlands Cancer Institute, Plesmanlaan 121, 1066 CX Amsterdam, The Netherlands

**Keywords:** advanced melanoma, survival, center variation

## Abstract

**Simple Summary:**

The survival of advanced melanoma patients has improved significantly over the last decade due to the introduction of new systemic therapies. It is unknown whether survival outcomes of advanced melanoma patients differ between melanoma centers in the Netherlands. This research aimed to assess center variation in treatments and 2-year survival probabilities of advanced melanoma patients diagnosed between 2013 and 2017 in the Netherlands. Significant center variation in 2-year survival probabilities of patients diagnosed in 2014–2015 was observed after correcting for case-mix and treatment with new systemic therapies. The different use of new systemic therapies partially explained the observed variation. From 2016 onwards, no significant difference in 2-year survival was observed between centers. This study shows the added value of quality monitoring with a national registry that enables the study of variation between centers.

**Abstract:**

Background: To assure a high quality of care for patients treated in Dutch melanoma centers, hospital variation in treatment patterns and outcomes is evaluated in the Dutch Melanoma Treatment Registry. The aim of this study was to assess center variation in treatments and 2-year survival probabilities of patients diagnosed between 2013 and 2017 in the Netherlands. Methods: We selected patients diagnosed between 2013 and 2017 with unresectable IIIC or stage IV melanoma, registered in the Dutch Melanoma Treatment Registry. Centers’ performance on 2-year survival was evaluated using Empirical Bayes estimates calculated in a random effects model. Treatment patterns of the centers with the lowest and highest estimates for 2-year survival were compared. Results: For patients diagnosed between 2014 and 2015, significant center variation in 2-year survival probabilities was observed even after correcting for case-mix and treatment with new systemic therapies. The different use of new systemic therapies partially explained the observed variation. From 2016 onwards, no significant difference in 2-year survival was observed between centers. Conclusion: Our data suggest that between 2014 and 2015, after correcting for patient case-mix, significant variation in 2-year survival probabilities between Dutch melanoma centers existed. The use of new systemic therapies could partially explain this variation. In 2013 and between 2016 and 2017, no significant variation between centers existed.

## 1. Introduction

The systemic treatment of metastatic melanoma has rapidly evolved over the last decade. Before 2011, unresectable stage IIIC and IV melanoma patients had few treatment options. However, since the introduction of the anti-CTLA-4 inhibitor ipilimumab (2011) [1], many new systemic treatment options have been approved (Appendix A). The approval of anti-PD-1 monoclonal antibodies (2015) [2,3], BRAF-inhibitors (2013, 2013, 2019), and MEK-inhibitors (2014, 2015, 2019) [4,5,6,7] and combination therapy of ipilimumab plus nivolumab [8] have resulted in improved overall survival (OS) of advanced melanoma patients over the last decade [9,10,11]. In the Netherlands, after approval by the EMA, an independent national committee has to approve the individual entities for reimbursement. This approval for reimbursement came in 2015 for nivolumab and pembrolizumab and in 2016 for dabrafenib in combination with trametinib and ipilimumab plus nivolumab.

The introduction of these new systemic treatments posed several challenges for healthcare specialists as well as policymakers. First, patient groups had to be identified that benefit most from the new systemic therapies. Secondly, these new therapies are expensive, and there were doubts about their cost-effectiveness. Finally, experience in recognizing and treating severe adverse events that occur in administering checkpoint inhibitors was initially limited.

To address these challenges, since 2012, every patient with unresectable stage IIIc or IV melanoma in the Netherlands has been treated in a dedicated melanoma center and registered in the Dutch Melanoma Treatment Registry (DMTR). The setup of this national registry was one of two conditions the minister of health dictated as a condition for approval of the implementation of the new therapies. The second condition was the centralization of the care of patients with advanced melanoma. Based on their expertise in treating advanced melanoma and their geographic location (Appendix A), fourteen melanoma centers were selected by the Dutch Society of Medical Oncologists (NVMO). Data from this nationwide registry are, amongst other purposes, used to develop quality indicators to assess hospital performance.

In surgery, the benchmarking of performance on quality indicators is common [12]. In other domains such as medical oncology, the benchmarking of performance between centers and individual oncologists is not yet widely implemented. The primary aim of this study was to assess center variation in survival up to two years after diagnosis with advanced melanoma in 2013–2017 in the Netherlands. The secondary aim was to investigate whether differences in treatment patterns may be related to survival outcomes.

## 2. Materials and Methods

### 2.1. Study Design and Population

This study used data from the Dutch Melanoma Treatment Registry (DMTR). Data are prospectively collected from diagnosis until death. Follow-up is checked every three months by a trained data manager for changes in disease status, patients or disease characteristics, and treatment. A detailed description of the DMTR setup has been published by Jochems et al. [13]. For this study, all patients aged ≥ 18 years and older, diagnosed between 1 January 2013 and 31 December 2017 with unresectable stage IIIC-IV melanoma, were included. Patients with uveal and mucosal melanoma were excluded from this analysis.

### 2.2. Melanoma Centers

All fourteen melanoma centers in the Netherlands were included in this study. Of these fourteen centers, seven centers are academic centers, and seven centers are general hospitals. Melanoma centers are spread across the country to provide patients with the best geographical coverage (Appendix A). Patients are assigned to the center where a medical oncologist first saw them.

### 2.3. Predictors of Overall Survival

Prognostic patient and tumor characteristics used for the adjustment of differences in case-mix between the 14 melanoma centers were based on previous studies [14,15]. All regression models included: age at diagnosis, gender (male, female), baseline Eastern Cooperative Oncology Group Performance Status (ECOG PS) (0–1, ≥2), baseline lactate dehydrogenase levels (LDH; normal, 250–500 U/L, >500 U/L), organs with distant metastasis (<3 organ sites, ≥3 organ sites involved), brain metastasis (none, asymptomatic, and symptomatic), liver metastasis (yes, no), and BRAFV600 mutational status (wild-type, mutant). 

### 2.4. Statistical Analysis

Baseline patient and disease characteristics were analyzed using descriptive statistics. OS was defined as the time from diagnosis with unresectable IIIc or stage IV disease until death from any cause. Patients alive or lost to follow-up were right-censored at the time of last registered contact. Patients with a follow-up longer than 2 years were artificially censored at 2 years. We chose 2-year survival as our primary outcome as 2-year follow-up was completed for almost all patients diagnosed between 2013 and 2017.

Cox proportional hazard models were used to assess the association of prognostic factors with survival probability during the first 2 years after diagnosis. Patient, tumor, and center characteristics were included in the first Cox multivariable regression model with a shared frailty (random effect) for center ID for up to 2-year survival probability to adjust for differences in patient case-mix between different centers (model I). In all models, we assumed that the frailty followed a gamma distribution. Shared frailty models attempt to separate true differences between centers from random fluctuation. These models yield “Empirical Bayes estimates” of the residual center effects, which were expressed as “center hazard ratios (HRs)” with respect to the national average [16]. To assess the influence of the use of new systemic therapies on differential outcomes in the centers, we extended the first model with first-line treatment with new systemic therapies (anti-PD-1 monotherapy, BRAF/MEK inhibitors, and/or ipilimumab plus nivolumab) as a patient characteristic (model II) [17]. In the final model (model III), we added a time-dependent covariate of new systemic therapies (anti-PD-1 monotherapy, BRAF/MEK inhibitors, and/or ipilimumab plus nivolumab) to model I. The three models used in this study are summarized below:I.Cox regression model for OS with patient and tumor characteristics as covariates and center ID modeled by a random effect;II.Cox regression model for OS with patient and tumor characteristics as covariates and center ID modeled by a random effect and baseline first-line systemic therapy as a patient characteristic;III.Cox regression model for OS with patient and tumor characteristics as covariates and center ID modeled by a random effect and new systemic therapies (anti-PD-1 monotherapy, BRAF/MEK inhibitors, and ipilimumab plus nivolumab) as a time-dependent covariate.

To determine which factors were associated with up to 2-year survival in all diagnosis years, we used model III. Treatment patterns of the three centers with the highest and lowest center HRs for patients diagnosed between 2013–2015 and 2016–2017 were compared using Sankey diagrams [18]. We chose these years as anti-PD-1, and the combination of BRAF/MEK inhibitors was available from 2015.

The degree of hospital variation between centers was expressed using median hazard ratios (mHRs). This concept has been described according to the methods by Austin et al. and can be interpreted as the median increase in hazard of mortality when comparing a patient at a hospital with higher mortality to a patient at a hospital with lower mortality [19]. The mHR was calculated and described by Austin et al., separately for each year of diagnosis for model I to III. Center variation was considered significant when the lower limit of the 95% confidence interval of the mHR was > 1.00.

Statistical software used was R [20] (version 4.0.1 ; packages car [21], tidyverse [22], survival [23], survminer [24], frailtyEM [25] and ggalluvial [26]).

## 3. Results

Between 2013 and 2017, a total of 3820 patients were diagnosed with unresectable stage IIIC/IV melanoma (Appendix A). Center size ranged from 110 in the smallest center to 955 patients diagnosed between 2013 and 2017 in the largest center. Baseline patient and tumor characteristics are shown in Table 1. Overall, patients had a median age of 64, ECOG PS 0–1 (84.8%), stage IV-M1c disease (76.7%), elevated LDH (36.8%), brain metastases (28.4%), liver metastases (29.1%), BRAFV600 mutation (45.6%), and metastases in ≥3 organ sites (46.4%). Patient and tumor characteristics of patients diagnosed between 2013 and 2017 for each center separately are shown in Appendix A, and the number of patients is shown in Appendix A. Median follow-up of all patients was 13.3 months (95%CI 12.5–14.2).

### 3.1. Factors Associated with 2-Year Survival

To determine which factors were associated with up to 2-year survival, we used a multivariable shared frailty Cox model (model III). The results show that each year of age (HR = 1.02, 95%CI: 1.01–1.02), ECOG PS ≥2 (HR = 1.91, 95%CI: 1.71–2.14), elevated LDH (250–500 U/L HR = 1.33, 95%CI: 1.20–1.47 and >500 U/L HR = 2.33, 95%CI: 2.06–2.64), brain metastases (asymptomatic HR = 1.44, 95%CI: 1.26–1.65 and symptomatic HR = 1.82, 95%CI: 1.65–2.01), liver metastases (HR = 1.31, 95%CI: 1.18–1.44), number of organ sites ≥3 (HR = 1.50, 95%CI: 1.37–1.65), and BRAFV600 wild-type (HR = 1.16, 95%CI: 1.07–1.26) were significantly associated with higher risk of death during the first 2 years since diagnosis (Appendix A).

### 3.2. Hospital Variation

Between 2013 and 2017, 2-year survival probabilities increased from 29% to 44% for all centers together. Based on the mHR of model I to III, significant variation between centers in 2-year survival probabilities existed in 2014–2015, but there was no significant variation for patients diagnosed in 2013 and between 2016 and 2017 (Figure 1). Model-based center-specific HRs of the different centers corrected for case-mix covariates are shown in Figure 2. In the year with the largest variation (2014), between-center HRs ranged between 0.72 and 1.36 relative to the national average. Between 2013 and 2015, mHR was 1.13. Figure 3 shows the model III-based survival curves for a male 80-year-old reference patient with 1 × elevated LDH, an ECOG PS of ≥2 with 0–2 organ sites, no liver or brain metastases, and BRAFV600 wild-type tumor diagnosed in 2014 vs. the same patient diagnosed in 2016.

### 3.3. Treatment Patterns

Treatment patterns were different in the centers over the years due to the introduction of anti-PD-1, BRAF/MEK combination therapy, and combination therapy of ipilimumab plus nivolumab. Treatment patterns of the three centers with the lowest HRs (centers with the best outcome) and the highest HRs (centers with the worst outcome) for death until 2 years are shown in Figure 4 and Figure 5. Overall, the centers with the lowest HRs treated a higher percentage of patients with anti-PD-1 antibodies (31% vs. 20%) and BRAF/MEK inhibitors (19% vs. 5%) in the first three lines of treatment compared to centers with higher HRs between 2013 and 2015. Compared to the treatment patterns for patients diagnosed in 2013–2015, this is different for patients diagnosed in 2016–2017. Overall, a majority of patients diagnosed between 2016 and 2017 received anti-PD-1 antibodies (48% vs. 60%) and/or BRAF/MEK combination therapy (40% vs. 29%) and/or ipilimumab + nivolumab (26% vs. 7%) either in the first, second, or third line for centers with the highest HRs (lowest 2-year survival) vs. centers with the lowest HRs (highest 2-year survival), respectively.

### 3.4. Impact of Anti-PD-1 and BRAF/MEK Inhibitors on Survival Probabilities

Depending on the center, 3–29% of patients diagnosed between 2013 and 2015 were treated with first-line anti-PD-1, BRAF/MEK combination therapy, and/or ipilimumab plus nivolumab anywhere in the first three lines of systemic therapy. Treatment with these systemic therapies in the first line was significantly associated with a lower hazard of death until 2 years after diagnosis (model II with all years combined: HR = 0.53, 95%CI: 0.49–0.58, *p*-value < 0.001) over all years of diagnosis. The inclusion of treatment with new systemic therapies as a time-dependent variable was also significantly associated with a lower hazard of death until 2 years after diagnosis (model III with all years combined: HR = 0.66, 95%CI: 0.61–0.72, *p*-value < 0.001). 

## 4. Discussion

Based on a nationwide prospective registry, this report describes hospital variation in 2-year survival probabilities and treatment patterns of advanced melanoma patients between 2013 and 2017 in the Netherlands. Between 2013 and 2017, nationwide 2-year survival probabilities of advanced melanoma increased from 29% to 44%. The observed variation between centers was statistically significant after correcting for case-mix and treatment with new systemic therapies in 2014–2015 and was non-significant in 2013 and 2016–2017. Two-year survival probabilities were significantly associated with the first-line treatment of anti-PD-1, BRAF/MEK inhibitors, and/or ipilimumab plus nivolumab (HR = 0.62, 95%CI: 0.54–0.73, *p*-value < 0.001), with higher 2-year survival probabilities in centers that treated more patients in the first line with these therapies. Between 2013 and 2015, mHR was 1.13.

Until now, no study has described that hospital variation in survival outcomes for advanced melanoma patients may be related to differences in treatment patterns between centers. This study shows the added value of quality monitoring with a national registry that enables the study of variation between centers. Despite the constantly changing therapeutic landscape between 2013 and 2017, Dutch melanoma centers have ensured that, currently, no significant treatment and 2-year survival variation between centers exists. This may be attributed to several reasons. Firstly, in 2012, Dutch melanoma care was centralized to fourteen centers. This resulted in the establishment of more high-volume melanoma centers. A previous study has shown that for patients with advanced melanoma, treatment in high-volume hospitals (*n* > 10) was associated with improved OS [27]. This study does not observe such an effect, possibly by the minimum volume of twenty newly diagnosed patients each year. Secondly, the fourteen melanoma centers meet quarterly in scientific meetings, sharing new developments and best practices. This may have enabled the quick transmission of knowledge and practice across the different centers.

The current study shows that center variation in 2-year survival probabilities was only significant in patients diagnosed between 2014 and 2015. Approval and reimbursement of these new systemic therapies happened mainly in 2014–2015. However, some centers had access to these treatments before an official decision on reimbursement by participation in trials, compassionate use, or extended access programs, which might explain the variation in 2-year survival probabilities. The bar plots presented in this study show this effect, as the best performing centers have higher percentages of these new treatments in the first, second, and third line of anti-PD-1 antibodies (31% vs. 20%) and BRAF/MEK inhibitors (19% vs. 5%) between 2013 and 2015 (Figure 5). Over the years, this inequality in access has disappeared, resulting in complete access to these systemic treatment options in every melanoma center in the Netherlands. However, the percentage of patients treated with combination therapy of ipilimumab plus nivolumab remains different between the centers with the lowest and highest HRs (26% vs. 7%), but this does not lead to significant differences in 2-year survival. Future research should focus on current differences between centers in using combination therapy of ipilimumab plus nivolumab.

A previous real-world study of treatment patterns in the USA using observational data shows that about 15% of all patients received first-line treatment with ipilimumab in 2016 [28]. In comparison, in our study, we observe almost no patients diagnosed between 2016 and 2017 treated with ipilimumab monotherapy as a first-line treatment across all centers in the Netherlands. This can be explained by the study of Robert et al., who showed that treatment with pembrolizumab resulted in a better OS as compared to the treatment with ipilimumab [2], or by Larkin et al. [29], who showed that combination treatment with nivolumab and ipilimumab resulted in significantly better OS compared to ipilimumab monotherapy. Rates of patients receiving anti-PD-1, BRAF/MEK-inhibitors, and combination therapy of ipilimumab plus nivolumab are comparable across both studies. However, compared to randomized clinical trials, 2-year survival probabilities in this cohort study are lower [7,30], possibly due to the inclusion of older patients, more patients with stage IV-M1c disease, and a higher ECOG PS of ≥2 [31]. 

The limitation of this study is that Cox models with a frailty component shrink the estimates compared to crude estimates. These models allow for the separation of observed center effects into explained variation by case-mix and center characteristics, unexplained differences between center outcomes, and random fluctuations [32]. Using these center-specific estimates helps in the interpretation of outcomes of small centers and prevents over-estimation of the effects. However, as a downside of using these frailties, the effects of some centers may be underestimated, and differences between the worst and best-performing centers may even be larger than presented in this study [17]. 

Based on the presented results, significant hospital variation in 2-year OS of advanced melanoma patients existed in 2014–2015 in the Netherlands. This significant hospital variation can partially be explained by case-mix and differences in percentages of patients treated with anti-PD-1, BRAF/MEK combination therapy, and combination therapy of ipilimumab and nivolumab. From 2016 onwards, no significant hospital variation corrected for case-mix was found, possibly due to similar treatment patterns across all centers, as a result of the nationwide collaboration of medical oncologists in the DMTR. Currently, the two-year survival probabilities of advanced melanoma patients in the Netherlands are comparable and do not differ between centers. However, in countries with no quality assurance in the form of centralization and a national registry with feedback, variations in outcomes between hospitals treating advanced melanoma patients might still be present.

## 5. Conclusions

In conclusion, we show significant differences in 2-year survival between Dutch melanoma centers between 2014 and 2015. The different pace of implementation of new cancer treatment of metastatic melanoma affected survival outcomes, but this difference disappeared in later years. To improve overall survival, new registered cancer treatments must be implemented by collaboration between medical professionals, regulatory authorities, and the organization within hospitals.

## Figures and Tables

**Figure 1 cancers-13-05077-f001:**
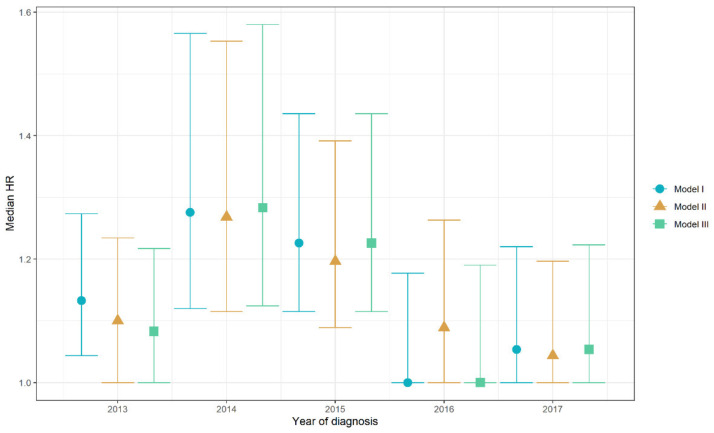
Hospital variation in 2-year survival probabilities expressed as mHRs in three Cox frailty models with gender, age at diagnosis, ECOG PS, LDH levels, number of organ sites, liver metastases, brain metastases, and BRAF mutational status as covariates and year of diagnosis as a factor. Intervals represent 95% confidence intervals of the mHRs. A mHR of 1.00 means that no center variation is present. The larger the mHRs, the larger the variation in 2-year probabilities between centers. The figure shows that significant variation in 2-year survival probabilities was significant between 2013 and 2015 and not significant in 2016 and 2017.

**Figure 2 cancers-13-05077-f002:**
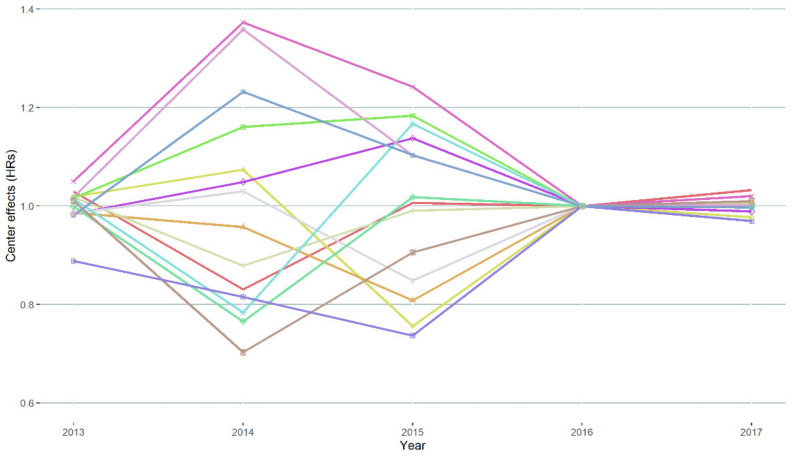
HR for death until 2 years of each center between 2013 and 2017 in model III. Higher ratios mean that a center has a lower observed/expected ratio for 2-year survival corrected for gender, age at diagnosis, ECOG PS, LDH levels, number of organ sites, liver metastases, brain metastases, and BRAFV600 mutational status as covariates in a specific year of diagnosis. HRs = hazard ratios.

**Figure 3 cancers-13-05077-f003:**
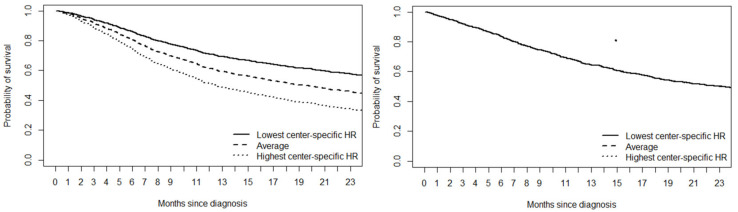
Model III-based predicted survival curves of patients diagnosed in 2014 (left) and 2016 (right) with the following characteristics: 80-year-old male, with 1 × elevated LDH, ECOG PS of ≥2 with 0–2 organ sites, no liver or brain metastases, BRAF wild-type tumor, who would be treated in the centers with the highest, lowest, and average HR in the dataset. HR = hazard ratio.

**Figure 4 cancers-13-05077-f004:**
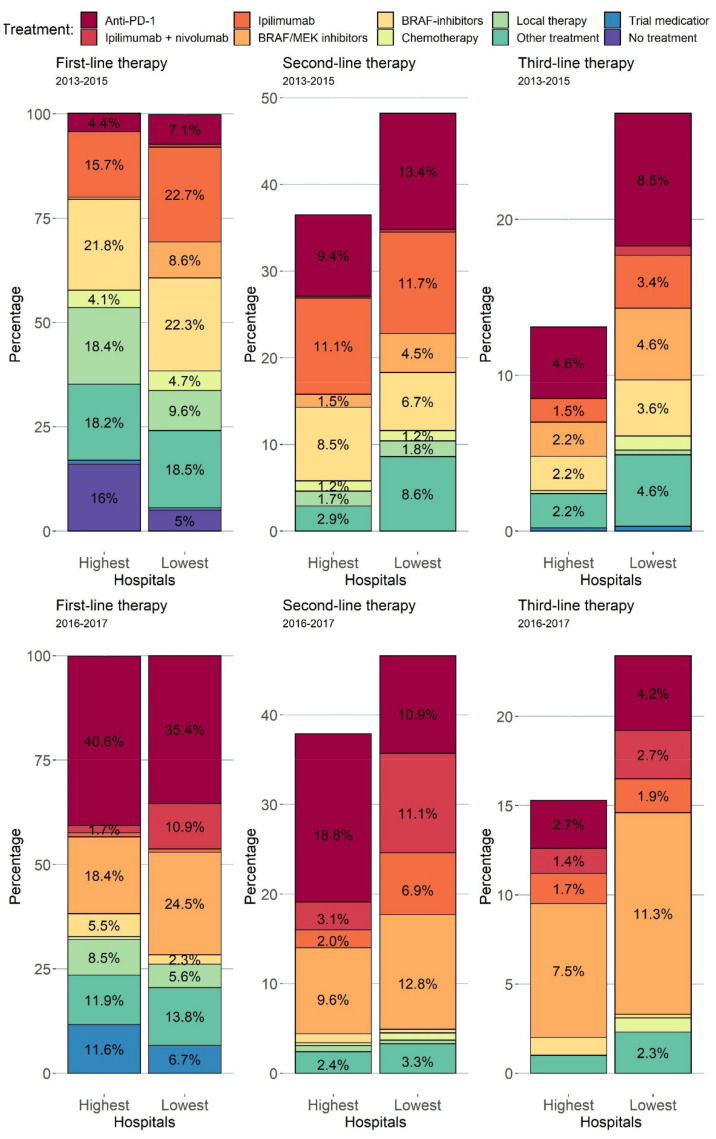
Stacked bar plots showing the treatments in the first, second, and third line of treatment. The three hospitals with the highest HRs (*N* = 413) and lowest 2-year survival are compared to the three hospitals with the lowest HRs (*N* = 674) according to the model. The top row shows patients diagnosed between 2013 and 2015, and the bottom row shows patients diagnosed between 2016 and 2017. Treatments administered after 3 December 2020 and labels of percentages < 1% are not shown in this barplot.

**Figure 5 cancers-13-05077-f005:**
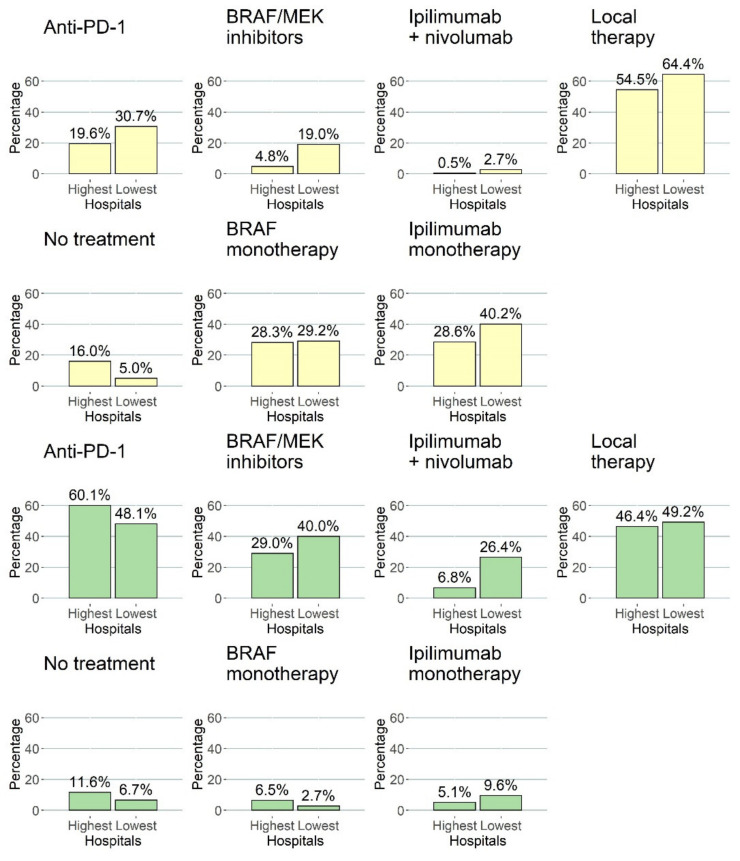
Bar plots showing the percentage of patients with a specific systemic therapy somewhere in their treatment after diagnosis with advanced melanoma. Yellow bars represent patients diagnosed between 2013–2015, and the green bar plots show patients diagnosed between 2016 and 2017. Columns depicted by “highest” are the three centers with the highest HRs for 2-year survival, and “lowest” are the centers with the lowest HRs for 2-year survival.

**Table 1 cancers-13-05077-t001:** Patient and tumor characteristics of patients diagnosed between 2013 and 2017. If ≥5% of data are missing for a variable, this is indicated by the category “unknown”.

Variable	Level	Total (*N*) = 3820
Age (median (range))		64 (18–97)
Gender (%)	Male	2254 (59.0)
	Female	1565 (41.0)
ECOG PS (%)	0–1	2832 (84.8)
	≥2	507 (15.2)
	Unknown	481 (12.6)
Stage (%)	IIIc unresectable	190 (5.0)
	IV-M1a	297 (7.8)
	IV-M1b	397 (10.5)
	IV-M1c	2903 (76.7)
LDH (%)	Normal	2215 (63.1)
	1–2 × ULN	816 (23.2)
	>2 × ULN	479 (13.6)
	Unknown/not determined	310 (8.1)
Brain metastases (%)	No	2672 (71.5)
	Yes, asymptomatic	334 (8.9)
	Yes, symptomatic	730 (19.5)
Liver metastases (%)	No	2667 (70.9)
	Yes	1094 (29.1)
Organ sites (%)	<3	2047 (53.6)
	≥3	1773 (46.4)
BRAF^V600^ mutation (%)	Wild-type	1741 (45.6)
	Mutant	2079 (54.4)

## Data Availability

The datasets generated during and/or analyzed during the current study are not publicly available due to privacy regulations in the Netherlands but are available from the corresponding author on reasonable request.

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
