# Peer review of "Hospital Variation in Cancer Treatments and Survival OutComes of Advanced Melanoma Patients: Nationwide Quality Assurance in The Netherlands"

_cancers, 2021, doi:10.3390/cancers13205077_

Round 1

Reviewer 1 Report

The submitted manuscript is sounding and well organized, aimed to evaluate the evolution of melanoma treatments in Netherland.

The results are interesting and also stress the importance of continuative knowledge sharing thanks to audit series.

Very interesting the use of the R software package for statistical analysis.

I suggest to revise the text for typos and to add the means of some acronyms (i.e., OS, HR, etc) when used the first time.

At last, fig. 6 is not clear and should be more readable

Reviewer 2 Report

The authors have described in this article epidemiological studies in 3820 melanoma patients. The patients were enrolled in 14 Dutch centers. The authors have been studied the correlation between melanoma cancer at III and IV stages and three variables pointing out three different models.

In my opinion, the study has statistically relevance about the first-frontline treatments. Minor point must be highlighted to improve the manuscript. 

Minor points.

Q1. The authors must describe the correlation with liber and brain metastases and the gender. Did they find any association with the onset of secondary metastasis and the patients gender? Did it sound statistically relevance?

Q2. Did the authors find a decrease in brain and liver metastasis upon first-line treatments ad described in the article in 2013-2015 periods?

Q3. How the authors explain the failure of therapeutics treatments in the subsequents years?

Q4. The authors should describe in Material and Methods the solvent used to dissolve therapeutics used in this epidemiological study. It is know that several solvent induced adverse effect in cancer patients. Have the authors any opinion about it?

This information is necessary to elicit side effects that it could be associated to the onset of secondary cancers or metastasis. 

Q5. The authors must highlight the BRAF mutation in more or less of 50% melanoma patients and the onset and progression of this cancer. How do they explain it?

Q6. The authors must improve English in Discussion Section.
